



# Experimental tests of phytoplankton response to ornithological eutrophication in Arctic freshwaters

Heather L. Mariash[1], Milla Rautio[2], Mark Mallory[3], Paul A. Smith[1]

[1]National Wildlife Research Centre, Science and Technology Division, Environment and Climate Change Canada, Ottawa,
Ontario, K1A 0H3, Canada.
[2]Centre d'études nordiques et Département des sciences fondamentales, Université du Québec à Chicoutimi, Chicoutimi,
Québec, Canada
[3]Biology, Acadia University, Wolfville, Nova Scotia, B4P 2R6, Canada,

10 *Correspondence to*: Heather L. Mariash (heather.mariash@gmail.com)

**Abstract.** Many populations of Arctic-breeding geese have increased in abundance in recent decades, and in the Canadian Arctic, Snow (Chen caerulescens) and Ross' Geese (Chen rossii) are formally considered overabundant by wildlife managers. The impacts of these overabundant geese on terrestrial habitats are well documented, and more recently, studies have suggested impacts to freshwater ecosystems as well. The direct contribution of nutrients from goose faeces to water chemistry could have

15 cascading effects on biological functioning, through changes in phytoplankton productivity and community composition. We demonstrated previously that goose faeces can enrich ponds with nutrients at a landscape scale. Here, we show experimentally that goose droppings rapidly released nitrogen and phosphorus when submerged in freshwater, increasing the dissolved nitrogen and phosphorus in the water. This resulted in both a decrease in the nitrogen:phosphorus ratio and an increase in cyanobacteria in the goose dropping treatment. In contrast, this pattern was not found when we submerged cut sedge (Carex

20 sp.) leaves. These results demonstrate that geese act as biovectors, causing terrestrial nutrients to be bioavailable in freshwater systems. Collectively, the results demonstrate the direct ecological consequences of ornithological nutrient loading from hyperabundant geese in Arctic freshwater ecosystems.



## 1 Introduction

Arctic regions are important breeding grounds for a wide range of migratory species. With 22.1 million geese, belonging to five species, breeding in the Canadian Arctic (Fox and Leafloor, 2018), along with substantial numbers of non-breeders, geese are ubiquitous on the Arctic landscape during the summer. Goose populations have been increasing since the 1950s, primarily

due to changes in agricultural practices that have increased food availability in the southern wintering grounds, but also because of increased survival from increased use of wildlife reserves and protected areas, as well as milder winters (Abraham et al., 2005).

Increases in abundance have been especially pronounced for several populations of Snow (Chen caerulescens) and Ross's

Geese (Chen rossii) (Fox and Leafloor, 2018). These large and increasing populations have caused considerable change to the Arctic habitats that they use for staging, breeding and brood-rearing (Abraham et al., 2012). Geese provide both deleterious and beneficial ecosystem services to tundra habitats (Buij et al. 2017). In a negative role, repeated overgrazing of graminoid forage plants weakens them, and grubbing of the below-ground plant parts compromises vegetation regrowth and the stability of pond edges (Jefferies et al., 2006). However, geese also rapidly liberate nutrients in an otherwise nutrient-poor landscape.

Because geese digest only a fraction of the plant material they ingest, they compensate with a high turn-over from feeding to faeces (Cadieux et al., 2005). This nutrient enrichment of the terrestrial environment can lead to enhanced primary productivity (reviewed in Buij et al. 2017, but see Gauthier et al. 1995). Geese predominantly graze around ponds, especially with broods and when they are moulting their flight feathers; the ponds are essential to escape predation. As a result, pond perimeters in areas used heavily by geese are notably mossy, brown and muddy due to the heavy localized grazing. At these pond margins

and indeed throughout the catchment, geese have the potential to influence freshwater ecosystems indirectly through this mobilization of nutrients.

While considerable research has outlined the effects of grazing and grubbing on terrestrial habitats, very few studies have focused on the associated freshwater habitats. Shallow freshwaters are highly connected to their catchments by a high surface

area to volume ratio (Rautio et al., 2011). Thus, ponds are very susceptible to habitat changes within the catchment, from increased terrestrial organic matter flowing into the pond due to heavily grazed pond edges, through the decomposition of goose droppings, and through sediment bioturbation that brings nutrients back into suspension. Increased terrestrial organic matter leads to a more bacteria-based production rather than photosynthetic (Ask et al., 2009). On the other hand, increased nutrients can cause shifts in trophic status and increased phytoplankton productivity, as demonstrated in nutrient addition

experiments (Schindler et al., 2008), or in systems where seabirds act as biovectors transferring marine nutrients to ponds (Michelutti et al., 2009). Arctic-breeding geese could be acting as biovectors, causing an accumulation of terrestrial nutrients in ponds, but so far there is little published information on rates of nutrient loading into freshwater systems in the Arctic (Dessborn et al., 2016).



Additions of nitrogen and phosphorus in temperate waters can clearly contribute to eutrophication, resulting in pronounced shifts in community composition and ecosystem function (Pace et al., 2010; Schindler et al., 2008). While it has been demonstrated that goose droppings are a significant source of total nitrogen (N) and phosphorus (P) to nearby ponds (Côté et al., 2010; Mariash et al., 2018; Olson et al., 2005), only a few studies have attempted to quantify the magnitude of ornithological nutrient loading (Liu et al., 2014; Post et al., 1998), and relate increased nutrients to broader limnological affects (Van Geest et al., 2007; MacDonald et al., 2015; Unckless and Makarewicz, 2007). Currently, almost no measures of these broader ecosystem-level effects exist for Arctic ponds.

Arctic ponds, typically characterized as oligotrophic transparent waters, are increasingly represented by turbid, mesotrophic waterbodies (Wauthy et al., 2018; Wrona et al., 2016). Increasing Arctic temperatures, and increased permafrost thaw play an important role in these changes (Vonk et al., 2015; Wauthy et al., 2018). However, geese are another potential vector of change in these ecosystems through direct (faeces) and indirect (bioturbation) nutrient release. Arctic freshwater wetlands used by geese are vital feeding and breeding grounds for many migratory bird species, including many sympatric and declining species of shorebirds (Flemming et al., 2019). Given the crucial ecological role played by freshwater wetlands in the Arctic, an understanding of goose-related habitat change in Arctic freshwaters has been identified as a research priority by goose population managers.

To better understand how ornithological nutrient loading affects both water chemistry and biological functioning, we designed a study to: i) measure the nutrients released from submerged goose droppings, ii) measure the concentration of dissolved nutrients in the water over time in the presence of these droppings, and iii) measure the resultant changes to phytoplankton productivity and community composition. We hypothesized that goose droppings would increase nutrient loading in the water, specifically total nitrogen and phosphorus, and that this increase in nutrients would increase primary production and change phytoplankton community composition, with a shift towards increased presence of cyanobacteria.

## 2 Methods

### 2.1 Mesocosm experiment

We measured the primary production and phytoplankton community composition response to nutrients released into the water column by submerged goose droppings and *Carex*, using an *in situ* mesocosm experimental approach. In order to differentiate between nutrients released from the plant detritus and nutrients cycled through geese, we compared the nutrients released from undigested graminoid clippings, goose droppings, and pond water only treatments. The *in situ* mesocosm experiment was established in the wetlands of Southampton Island, Nunavut, Canada, at the East Bay long-term field station (63°59'N,



81˚40'W). Fresh goose droppings containing both faeces and uric acid from lesser snow geese and fresh clippings of the dominant graminoid (*Carex sp.*), herein simply referred to as *Carex*, were all collected on 8 July 2015. The goose droppings were all less than 24 h old, moist and green in color; the droppings were pooled and homogenized, while the *Carex*, including the bulb and blade, was clipped into 2 cm long pieces. We placed $10 \pm 0.05$ g of goose droppings or $4$ g $\pm 0.05$ g of fresh *Carex*

(approximately equivalent to 1.1 g dry mass), into plastic cups, which were filled with 200 ml of pond water, that had been passed through a $< 50$ μm sieve. The cups were placed in a floating wooden frame to keep each container upright and floating in the pond, in order to retain some natural turbulence (Fig 1). The cups were covered with plastic wrap to allow light through but prevent evaporation or overfilling. The experiment was conducted for 17 days in the pond at ambient temperatures (average temperature in the cups 8.2 °C) and natural light conditions (approximately 21 h daylight, 3 h of dusk).

Treatments were goose droppings and *Carex* with five replicates of each treatment for each sampling day, along with control cups containing only pond water. Water parameters were sampled on day 1, 3, 5, 10, and 17. On sampling days, we sampled both the overlying water and the organic matter from the five cups per treatment to measure the rate of nutrients released into the water and decomposition of the *Carex* and goose droppings. For nutrient samples, the water from two of the five cups from

each treatment was passed through 50 μm sieve and poured into prepared vials for total phosphorous (40 mL volume with 116 μl of 30% $H_2SO_4$) and total nitrogen (24 mL volume with 230 μl HCL). The water from the remaining three replicates was pooled for phytoplankton community composition (150 ml amber glass bottle preserved with Lugol's iodine solution). Phytoplankton were only sampled during the first 10 days of the experiment, to limit the cup-effect on phytoplankton community dynamics.

## 2.2 Identifying an appropriate loading rate

To compare our results to those published from elsewhere, it was necessary to establish a "faecal loading rate" with consistent units. We define nutrient loading rate within the catchment as the concentration of nitrogen and phosphorus measured from goose droppings, scaled up using rates of defecation and the density of geese per km² reported for each site used in our

comparisons, for a final unit of kg of nutrients per km² per day. The loading rate for the region in which our study took place, Southampton Island, was calculated based on nutrient concentrations from the goose droppings used in the experiment, combined with a conservative estimate of faecal production based on weight and defecation rate per day (Unckless & Makarewicz, 2007) and the average density of geese breeding in Southampton Island goose colonies (Kerbes et al., 2014). We view this estimate as conservative because non-breeding geese are numerous across Southampton Island (perhaps

outnumbering breeding geese), goslings make a substantial contribution to dropping densities. Both Kitchell et. al. (1999) and Olsen et. al. (2005) used the daily excretion rate for N and P as reported in Post et. al. (1998). For these studies, values for Table 3 used the minimum levels reported in Post et. al. (1998), which were 0.001 kg N day$^{-1}$ and 0.0002 kg P day$^{-1}$ per goose.

## 2.3 Laboratory analysis





The particulate organic matter, goose droppings or *Carex*, from each cup was frozen in the field and later freeze dried and weighed, ground, then subsampled for carbon, nitrogen, and phosphorus content. Carbon and nitrogen content were analysed at the University of Ottawa's G.G. Hatch Isotope Laboratory, using an elemental analyser (Elementar Isotope Cube, Germany), from samples (*Carex* $2.5 \pm 0.4$ mg; goose droppings $4.5 \pm 0.5$ mg) and standards that were weighed into tin capsules and loaded

into the elemental analyser. The phosphorus from the solid samples was first processed by dissolving 0.1 g of each sample in 5 ml concentrated $HNO_3$ for 1 h at 95 ˚C, with an additional 1 ml of $H_2O_2$ 30% added to each sample then incubated for another 2 hr at 95 ˚C. Ultra-pure water was added to complete the sample volume to 50 ml. These samples were then analysed by inductively coupled plasma atomic emission spectroscopy (ICP-AES: Varian Vista AX, Palo Alto, California, USA). The amounts of C, N, and P are expressed relative to the initial amounts of nutrients (%;Table 1). For the water samples, total

dissolved nitrogen (TN) and total phosphorous (TP) were analysed using catalytic combustion with a Shimadzu VCPH (Kyoto, Japan), including 3 blanks of ultra-pure water, at the Institut National de la Recherche Scientifique Centre-Eau Terre Environment (INRS, Québec, Canada).

For the comparison of faecal loading rate and the subsequent effects on water chemistry at a landscape scale, we report
previously unpublished chlorophyll-a values from a large-scale survey of lakes across Southampton Island (for sampling details see Mariash et. al. 2018)(Table 3). Water samples from these lakes were filtered onto GF/F filters (in duplicates), frozen at -80˚C, and later analysed using a Cary Eclipse fluorescence spectrophotometer (Aglilent, Santa Clara, USA) using standardized extraction methods and calculations (Holm-Hansen and Riemann, 1978; Jeffrey et al., 1997).

**2.4 Phytoplankton productivity and community composition**

Phytoplankton biovolumes, volume of cells per volume of water, and community composition were measured from Lugol-preserved samples using Utermöhl settling chambers (Utermöhl, 1958), and an inverted phase contrast microscope (Zeiss Axio Observer.A, Germany). A minimum of 400 cells per sample were counted, using 400x magnification until 200 cells were counted and 100x magnification for the remaining 200 cells. This ensured that both larger and smaller cells were accounted

for. A minimum of 10 fields were counted with each magnification. Biovolume estimates were based on geometrical models and cell measurements using photography and the AxioVision software (Zeiss, Germany) and converted using carbon to volume relationships (Menden-Deuer and Lessard, 2000). Phytoplankton production was estimated from the change of biovolumes between different sampling days. Taxa were identified to genera when possible but later grouped by class for comparisons. One phytoplankton sample (150 ml) was taken for each sampling, Day 1, 3, 5, 10. Three samples had insufficient

preservation and could not be quantified.

**2.5 Data analysis**

A General Linear Model was used to test for differences in the rate of nutrient loss from our organic matter treatments (goose droppings or *Carex* clippings. Time (days), time$^2$, treatment and their interactions were included as predictors, and nutrients



(C, N, P, N:P ratio) measured in the organic matter were the response variables. A similar analysis was conducted for the corresponding changes to nutrient concentrations in the experimental water. To visualize results, we fit a quadratic function with a 95% confidence interval (CI) to the relationship between TP and TN concentrations in the water and time, throughout the experiment, using *ggplot2* (Wickham, 2009). To model the effect of our treatments on primary productivity, we used a

linear mixed effects model implemented in *lme4* (Bates et al., 2014). We entered categorical light level, treatments including control, time (days), and an interaction between treatment and time into the model as fixed effects. Replicates were included as a random effect. Visual inspection of residual plots did not reveal any obvious deviations from homoscedasticity or normality (Zuur et al., 2010). P-values were obtained by likelihood ratio tests. Phytoplankton biodiversity was calculated using the Shannon-Wiener (alpha diversity) index. All analyses were conducted using R software (R version 3.1.1; (R Core Team,

2016) , and all means are reported ± SD unless otherwise noted.

## 3 Results

### 3.1 Nutrients released from submerged organic matter

The initial composition of the solid material showed that goose droppings had higher water content compared to *Carex*, a significantly higher percentage of nitrogen and phosphorous, and a significantly lower content of carbon and N:P ratio (Table

1, 2a,b,c). Once submerged, the carbon remained largely intact, showing a small but statistically significant loss of around 2% for goose droppings and 5% for *Carex* (Fig. 2a, Table 2a). Losses of nitrogen and phosphorus were much more rapid for the goose dropping treatment. After only 1 day, the goose droppings had released 48% of the original nitrogen and 43% of the original phosphorous, with no additional significant release of nutrients throughout the rest of the 17d experiment (Fig. 2b,c). In contrast, there was no net loss of nitrogen or phosphorus from *Carex* over the experimental period (Fig. 2b,c). The N:P ratio

stayed between 8- 10 for the goose dropping treatment, while the N:P ratio for *Carex* fluctuated between 11 – 14 (Fig 2d). Despite the rapid loss of nutrients from the goose droppings, phosphorous remained higher in the goose droppings compared to *Carex* at the end of the experiment (Fig. 2).

### 3.2 Nutrients released into the water

The nutrients released from the organic matter caused reciprocal changes in water chemistry during the experiment. The dissolved TN and TP in the water of the goose dropping treatment were orders of magnitude higher than the concentrations found in the water of the *Carex* treatment (Fig. 3). By the end of the experiment (Day 17), the water in the goose treatment contained 108 mg L$^{-1}$ TN, while the water in the *Carex* treatment had 1.6 mg L$^{-1}$ TN, compared with 0.1 mg L$^{-1}$ of the pond water initially, or the 0.6 mg L$^{-1}$ as the average TN from Southampton lakes (Mariash et al., 2018). Similarly, for TP, higher

concentrations were found in the water of the goose treatment compared to the *Carex* treatment (16 mg L$^{-1}$ TP compared to 0.2 mg L$^{-1}$ TP, respectively). The initial pond water concentration was 0.002 mg L$^{-1}$ TP, compared to the average TP concentration for local ponds of 0.01 mg L$^{-1}$. For total dissolved nitrogen, there was a significant main effect of treatment



(Table 2d). There was also a significant interaction between treatment and time (and time$^2$; Table 2d, Fig. 3a), indicating a larger increase over time for the goose treatment. Total dissolved phosphorus showed similar patterns, with a significant treatment effect, and significant time$^2$ and time$^2$*treatment effects (Table 2e, Fig. 3b).

### 5  3.3 Phytoplankton biovolumes and composition

Each treatment contained diverse phytoplankton communities and showed increased productivity over the course of the experiment. The initial phytoplankton biovolume was low at only 0.1 mm$^3$ L$^{-1}$. For both treatments, phytoplankton production, based on biovolumes, increased during the experiment reaching $4.6 \pm 0.2$ mm$^3$ L$^{-1}$ by Day 10 (Fig. 4). However, the production was dominated by different phytoplankton classes between treatments. In the *Carex* treatment, phytoplankton growth on Day

1 had risen to 1.2 mm$^3$ L$^{-1}$, which was mainly from the production of chlorophytes (0.64 mm$^3$ L$^{-1}$; 50 % of total biovolume), picoplankton (0.21 mm$^3$ L$^{-1}$; 16%), and bacillariophytes (0.16 mm$^3$ L$^{-1}$; 13 %, diatoms) (Fig. 4b). In the goose dropping treatment, phytoplankton production had doubled the *Carex* treatment, to a total of 2.5 mm$^3$ L$^{-1}$ on Day 3 (Fig. 4a). There was no rise in phytoplankton production between Day 3 and Day 5, but production doubled again between Day 5 to Day 10 to 4.6 mm$^3$ L$^{-1}$. Dominant phytoplankton in the goose dropping treatment were mainly picoplankton accounting for 2.3 mm$^3$ L$^{-1}$

(92%), chlorophytes 0.1 mm$^3$ L$^{-1}$, and 0.06 mm$^3$ L$^{-1}$ Chrysophyceae on Day 3, but rapidly changed to cyanobacteria (75%), on Day 5 (Fig. 4b). Cyanobacteria was no longer the main class but remained high on Day 10. The goose treatment had less taxa than the *Carex* treatment, with only 10 taxa present, represented by 8 classes. The most abundant taxa were *Chlorella, Ochromonas, Aphanocapsa,* and *Gonyostomum*, the latter being attributed to nuisance algal blooms. For the *Carex* treatment, the phytoplankton community was diverse at Day 10, with 19 taxa, represented by 12 classes, but was clearly dominated by

*Chlamydomonas,* a green algae (Chlorophyceae; 50%) on Day 10 (Fig. 4b). Cyanobacteria were not observed in the phytoplankton community of the *Carex* treatment.

These differences in phytoplankton communities among treatments (were also confirmed with diversity indices. Initially, the pond water had a Shannon-Wiener index of 1.9 with 16 taxa present. This increased to 2.2 and 2.3 on Day 1 and 10, respectively

for the *Carex* treatment. For the goose droppings treatment, the Shannon-Wiener index declined to 1.8 on Day 3 and 5, then returned to 1.9 on Day 10.

### 4 DISCUSSION

Previous studies have suggested that geese can act as an important vector of nutrients (especially N and P) from terrestrial to

aquatic systems, and this nutrient transfer might be especially important in otherwise oligotrophic Arctic ponds (Dessborn et al. 2016). However, studies from the Arctic are rare (Mariash et al. 2018, Mallory et al. 2006), and none to date have assessed the ecosystem response of goose-related eutrophication by linking the nutrient levels to primary production and community



responses. Our experimental field trials in wetlands on Southampton Island, Nunavut, provided evidence that submerged goose droppings leach a significant amount of nitrogen and phosphorous, immediately elevating the nutrient concentrations in the water. These leached nutrients were bioavailable, rapidly increasing phytoplankton production, and altering community composition. Both treatments showed diverse phytoplankton communities, however only in the goose dropping treatment did

cyanobacteria become dominant. Collectively, these results demonstrate the direct ecological consequences of ornithogenic nutrient loading in Arctic freshwater ecosystems.

**4.1 Nutrients released from submerged organic matter into the water**

Once submerged, goose droppings released approximately 45% of the nitrogen and phosphorus that they contained

on the first day. In a previous study, Lui et al. (2014) demonstrated that most nutrients were released from goose droppings in the first 10 days, but they did not measure the nutrient concentrations in the water. Also, our mesocosm approach allowed for natural light, temperature, and some mixing, along with more comprehensive tracking of the nutrients released from the goose droppings into the water, and the resultant effects on primary producers. We showed that this rapid release of N and P from goose droppings resulted in a rapid increase in TN and TP concentrations in the water column. The nutrient concentrations in

the water continued to increase until a peak at Day 10, then concentrations of TN and TP showed signs of decreasing on Day 17, when nutrients were presumably assimilated by phytoplankton.

Our mesocosm approach demonstrated that under natural light and temperature conditions, the release of nutrients from *Carex* and goose droppings, and the final dissolved nutrient concentrations in the water column, were quite different. While the *Carex*

clippings themselves were relatively N-rich, measuring 18 mg N g$^{-1}$ (nearly half of the concentration of N in goose droppings), there was no net change in nitrogen in the water column from *Carex* during our experiment. This was similar to the nutrient release dynamics found in Lui et al (2014), where the *Carex* at 10 °C had an immobilization phase within the first 5 days. In contrast, submerged goose droppings had a peak of nutrient release in the first day, while much of the remaining nutrients in the goose droppings were retained over the next 17 days. This high retention of the remaining P, approximately 50% over 17d,

has also been observed in other experiments (Liu et al., 2014). These retained nutrients in residual organic matter can settle into the sediment (Unckless and Makarewicz, 2007), where they can build up and later be re-suspended by a strong wind event. Additionally, the differences in the fractions of labile and recalcitrant nutrients may play a crucial role in the degree to which the nutrients contained in droppings versus *Carex* are available to influence productivity. Both a quick release of half the nutrients and sedimentation of the other half of the nutrients, in combination with high evaporation and low flushing rates, in

Arctic wetlands may further concentrate the nutrients leading to late summer eutrophic conditions (Lewis et al., 2015; Mariash et al., 2018).

While the experiment was helpful to demonstrate the rate of nutrients leaching into the water, the concentrations of goose droppings used were higher than natural loading rates calculated for Southampton Island goose colonies. To demonstrate



loading rates on a landscape scale, we compared natural loading rates and water chemistry changes from studies in goose colonies (Table 3). These studies reported a wide range of nutrient concentrations arising from natural droppings into the environment and experimental additions. The highest reported natural loading rates were found in the southern wintering grounds, with 9-15 kg N km$^{-2}$ d$^{-1}$ and 0.9-1.5 kg P km$^{-2}$ d$^{-1}$. In comparison, Schindler et al. (2008) added on average 298 kg of

N and 24 kg of P to a small boreal lake annually for the first 6 years of their classic whole-lake eutrophication experiment; when converted to a daily load per area, this was smaller than nutrient loads produced by goose colonies. Nutrient loads from geese translated into increases in both TN and TP in the water along with high Chl-a (Table 3) for all studies compared. Clearly, after entering the waterbody, many factors affect nutrient dynamics in the water, including residence time, depth, stratification, and algae biomass (Anderson et al., 2017). Nonetheless, despite the differences in climate and hydrology in these studies,

dissolved TN and TP were highest where the nutrient loads were highest. Moreover, at the landscape level, these nutrient load changes related to geese were occurring at a much faster pace than water chemistry changes caused by climate variables (Mariash et al., 2018).

**4.2 Phytoplankton response**

Chlorophyll-a (Chl-a) concentrations in shallow freshwaters averaged 1.9 µg L$^{-1}$ across the circumpolar arctic (Rautio et al., 2011), concentrations in pristine ponds in southwestern Greenland were lower, at 0.5 µg L$^{-1}$ (Mariash et al., 2014), while ponds on Southampton Island were slightly above that circumpolar average with 2.2 µg L$^{-1}$ Chl-a. In more southerly temperate wetlands, with higher nutrient loads from geese, Chl-a is orders of magnitude higher, with concentrations between 27 - 800 µg L$^{-1}$ (Kitchell et al., 1999). We had expected phytoplankton production to be higher in the goose dropping treatment vs. the

*Carex* treatment; however, the phytoplankton biomass was similar between the two treatments by the end of Day 10, showing that the input of nutrients derived from either treatment was enough for a phytoplankton response in these oligotrophic waters. Temperature may be another important factor limiting productivity; given enough nutrients, both light and temperature can restrict phytoplankton growth (Fanesi et al., 2016). Also, high concentrations of submerged *Carex* would occur primarily only along pond margins during periods of inundation, such as during the spring freshet, or when shoots are pulled by grazing geese

to consume the starchy base of the leaves. Thus, nutrients released from *Carex* may be insignificant compared to goose droppings at a landscape scale.

Community composition of the phytoplankton clearly responded to increased nutrient availability in both treatments. Initially dominated by diatoms, the phytoplankton community of the *Carex* treatment changed to having a more diverse community

with the dominant class being green algae. The phytoplankton community in the goose dropping treatment initially had only 10 taxa present represented by 8 classes, but by Day 5 the community was dominated (98%) by cyanobacteria (*Aphanocapsa* and *Pseudanabaena*) and chrysophytes (*Ochomonas*). A similar pattern in dominance of cyanobacteria and cryophytes is consistent with results in other nutrient-enrichment studies (Paerl et al., 2016; Przytulska et al., 2017). Cyanobacteria can outcompete other taxa when both nitrogen and phosphorous concentrations are high (Paerl et al., 2016; Schindler et al., 2008).



High abundance of cyanobacteria will negatively affect species richness and diversity, as seen in the diminishing presence of other phytoplankton classes in our experiment.

N- limitation of algae growth can occur at TN:TP ratios < 20 (Findlay et al., 1994; Guildford and Hecky, 2000). In Schindler
et al. (2008), the presence of cyanobacteria peaked within weeks after reducing the TN:TP from 12 to 4. Goose droppings in this study had an N:P ratio of 10, and when placed in water in our experimental treatment, they lowered the TN:TP ratio to 5.4, demonstrating goose droppings have the potential to significantly alter the ambient nutrient balance. The relatively phosphorous-rich goose droppings can cause nitrogen limitation in freshwaters (Mariash et al., 2018; Post et al., 1998; Schindler et al., 2008). This is an environmental concern from a water quality perspective, because when N is limiting, $N_2$-
fixing cyanobacteria are competitively favoured (Guildford and Hecky, 2000; Schindler et al., 2008). In the wintering grounds with highest goose densities, TN:TP ratios of the waterbodies had a mean of 15 (Kitchell et al., 1999), indicating N- limitation. While the wetlands across Southampton Island have TN:TP ratios of approximately 30 (Mariash et al. 2018), on par with other shallow Arctic freshwaters (Rautio et al., 2011), there is an indication that these wetlands are becoming more N-limited with decreasing TN:TP ratios (Mariash et al., 2018).

Geese are very inefficient herbivores, excreting approximately 60% of their ingested nutrients (Kitchell et al. 1999), and these nutrients are quickly released into the aquatic environment as demonstrated by the rapid release of nutrients from the organic matter in our goose dropping treatment. Once released these nutrients are bioavailable, altering water chemistry and the phytoplankton communities of the watershed. On Southampton Island, graminoids such as *Carex* spp. are the primary diet
source for the geese, and our experiment demonstrates that only when passed through geese was the nitrogen and phosphorus bound in the *Carex* released into the water. Geese are therefore acting as biovectors on the landscape, transforming large amounts of terrestrial nutrients into bioavailable nutrients in the freshwater ecosystems. As geese are long distance migrants, and as many circumpolar Arctic goose populations have increased substantially (Fox and Leafloor, 2018), their movement and effects on the aquatic habitats have implications across Arctic North America and Europe, at locations where geese congregate
in large numbers. Management strategies for hyper-abundant geese currently emphasize the need for maintaining the ecological integrity of terrestrial habitats. Our results demonstrate that the impacts of geese extend to freshwater Arctic ecosystems, and future management strategies should better acknowledge these aquatic impacts.

**Data archive statement:** Once accepted, the data will be made freely available on the Government of Canada's OpenData
system.





**Acknowledgements**

Financial support for data collection and analyses were provided in part by the Arctic Goose Joint Venture, the Canadian Wildlife Service, the Wildlife Research Division of Environment and Climate Change Canada, Canada Research Chair Program, and the Polar Continental Shelf Program. H.L.M was supported by a W. Garfield Weston Fellowship for Northern
5    Studies.





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





**Tables and Figures**

**Table 1. Initial composition of *Carex* sp. leaves and goose droppings, and the pond water used in the experiment. *Percent water is calculated from wet weight versus dry weight, while the other parameters are calculated as a percentage of dry weight (DW). Solid material had 5 replicates, while water chemistry assays were run in duplicates.**

| Material | *Carex* clippings Mean % | SD | Goose Droppings Mean % | SD | Pond water Mean mg L⁻¹ | SD |
|---|---|---|---|---|---|---|
| Percent Water* | 71.7 | 0.0 | 87.9 | 0.0 | | |
| Carbon | 46.6 | 0.4 | 43.6 | 0.3 | | |
| Nitrogen | 1.8 | 0.4 | 4.0 | 0.3 | 0.13 | 0.0 |
| Phosphorus | 0.1 | 0.4 | 0.4 | 0.1 | 0.002 | 0.0 |



**Table 2. Summary of results from General Linear Models comparing a) carbon b) nitrogen and c) phosphorus between the solid organic matter of *Carex* and goose dropping treatments and d) total dissolved nitrogen e) total dissolved phosphorous in the overlaying water. Significant values are in bold.**

|  | Source | Degrees of Freedom | MS | F | p-value |
|---|---|---|---|---|---|
| a) Carbon | treatment | 1 | 4543.1 | 26.4 | **< 0.001** |
|  | time | 1 | 837.4 | 4.9 | **0.03** |
|  | time^2 | 1 | 0 | 0.01 | 0.99 |
|  | treatment:time | 1 | 6 | 0.03 | 0.85 |
|  | treatment:time^2 | 1 | 240.3 | 1.3 | 0.244 |
|  | Residuals | 54 | 173.8 |  |  |
|  |  |  |  |  |  |
| b) Nitrogen | intercept | 1 |  |  |  |
|  | treatment | 1 | 548 | 20.58 | **< 0.001** |
|  | time | 1 | 66.6 | 2.5 | 0.11 |
|  | time^2 | 1 | 234.14 | 8.7 | **0.004** |
|  | treatment:time | 1 | 194 | 7.2 | **0.008** |
|  | treatment:time^2 | 1 | 270 | 10.1 | **0.002** |
|  | Residuals | 54 | 26.62 |  |  |
|  |  |  |  |  |  |
| c) Phosphorus | treatment | 1 | 30.1 | 85.9 | **< 0.001** |
|  | time | 1 | 0.17 | 0.38 | 0.49 |
|  | time^2 | 1 | 1.6 | 4.7 | **0.03** |
|  | treatment:time | 1 | 0.67 | 1.9 | 0.17 |
|  | treatment:time^2 | 1 | 3.9 | 11.2 | **0.001** |
|  | Residuals | 54 | 0.35 |  |  |
|  |  |  |  |  |  |
| d) TN dissolved | intercept |  |  |  | 0.953 |
|  | treatment | 2 | 35553 | 107.3 | **< 0.001** |
|  | time | 1 | 3572 | 21.6 | **< 0.001** |
|  | time^2 | 1 | 2162 | 13.1 | **0.001** |
|  | treatment:time | 2 | 5749 | 17.4 | **< 0.001** |





| | | | | | |
|---|---|---|---|---|---|
| | treatment:time^2 | 1 | 2185 | 13.2 | **0.002** |
| | Residuals | 20 | 3312 | | |
| e) TP | | | | | |
| dissolved | treatment | 2 | 616.6 | 59.6 | **< 0.001** |
| | time | 1 | 49.7 | 4.8 | **0.04** |
| | time^2 | 1 | 92.5 | 8.9 | **0.007** |
| | treatment:time | 2 | 39.5 | 3.8 | **0.04** |
| | treatment:time^2 | 1 | 94.6 | 9.1 | **0.007** |
| | Residuals | 20 | 10.4 | | |

**Table 3. Comparison of the loading rates of nitrogen (N) and phosphorus (P) from geese and these nutrients in dissolved form (TN, TP) in the water along with the Chlorophyll-a (Chl.a) concentrations found in the waterbodies. Loading rates are in kg nutrients per the density of geese per km² at a given site per day.**

| | | Nutrient Load | | | Water | | | |
|---|---|---|---|---|---|---|---|---|
| | | N | P | N:P | TN | TP | TN:TP | Chl.a |
| | | kg km$^{-2}$ | kg km$^{-2}$ | | mg | | | |
| Location | Study | day$^{-1}$ | day$^{-1}$ | | L$^{-1}$ | mg L$^{-1}$ | | μg l-1 |
| Rio Grande River, USA | Kitchell et. al. 1999 | 15.68 | 1.52 | 10.3 | 35.00 | 2.50 | 15.0 | 800.0 |
| Middle Creek Reservoir, USA | Olson et. al. 2005 | 8.90 | 0.86 | 10.3 | 4.00 | 0.08 | 53.3 | 94.3 |
| Southampton Island, Canada | Mariash et. al. 2018 | 4.46 | 0.50 | 8.9 | 0.45 | 0.02 | 30.1 | 2.2 |
| Ontario, Canada | Schindler et. al. 2008 | 2.72 | 0.22 | 12.4 | 0.83 | 0.04 | 20.8 | 27.0 |

High Bird density loads are reported from Kitchell et. al. 1999. For Olson 2005, the average nutrient loads for inflow and outflow are reported. For Southampton Island (SHI), values are the average of the 26 shallow waterbodies surveyed in 2015 (see Mariash et. al. 2018 for survey details). Schindler et. al. 2008 values are the average nutrient addition from the first 6 years when TN and TP were added with a similar ratio to the ratio found in goose droppings (after year 6, only P was added).





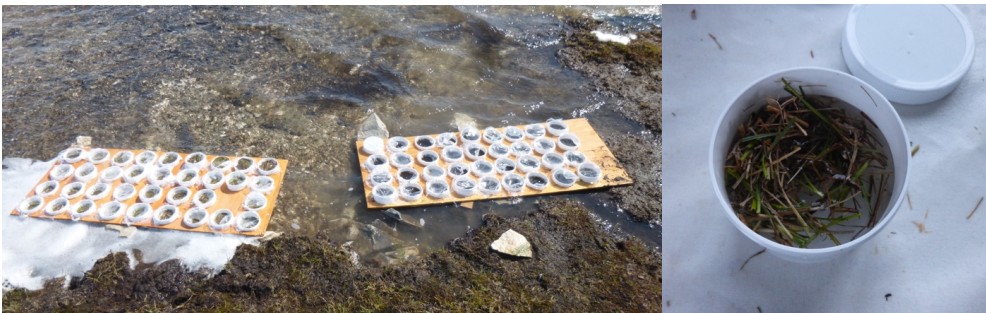

**Figure 1. Mesocosm set up in the pond, sample cups had either fresh *Carex* clippings or fresh goose droppings submerged in 200 mL of pond water. The organic matter and the water from experimental cups were sampled throughout the 17-day experiment.**

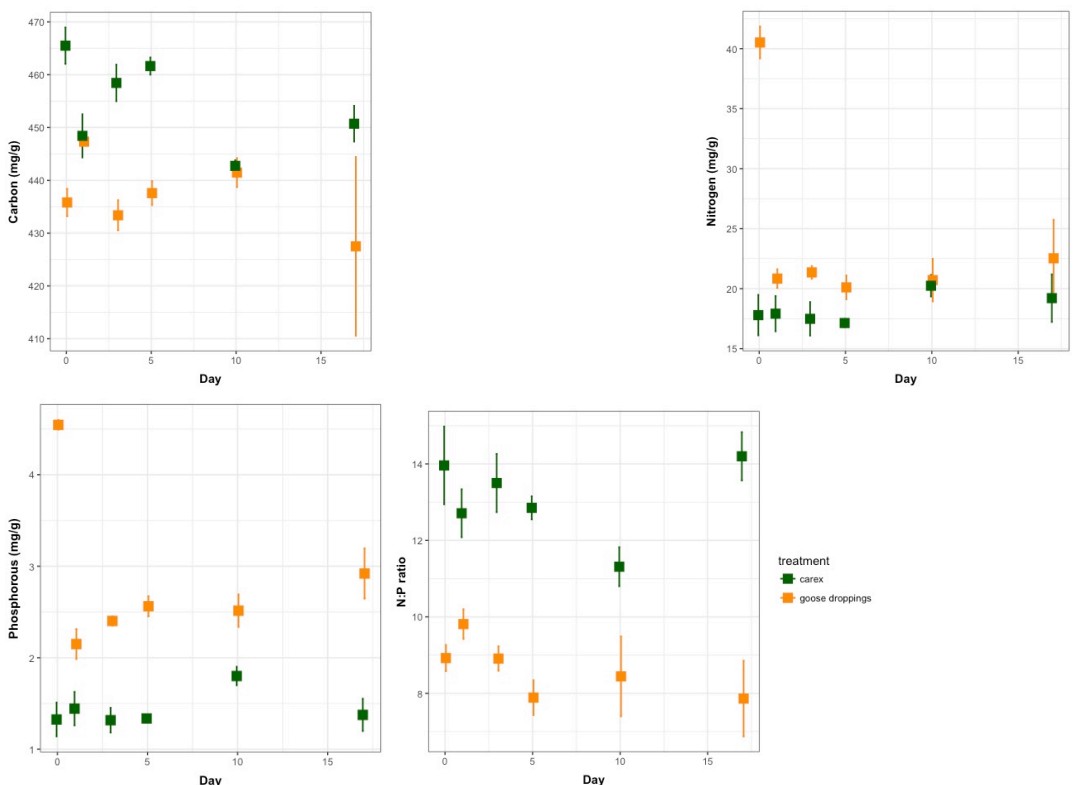

**Fig 2. The decomposition of a) carbon, b) nitrogen, c) phosphorus and d) nitrogen:phosphorous ratio, from submerged organic**
5 **material of *Carex* clippings and goose droppings, reported as mean ± SE across replicated (n=5) throughout the 17-day experiment.**




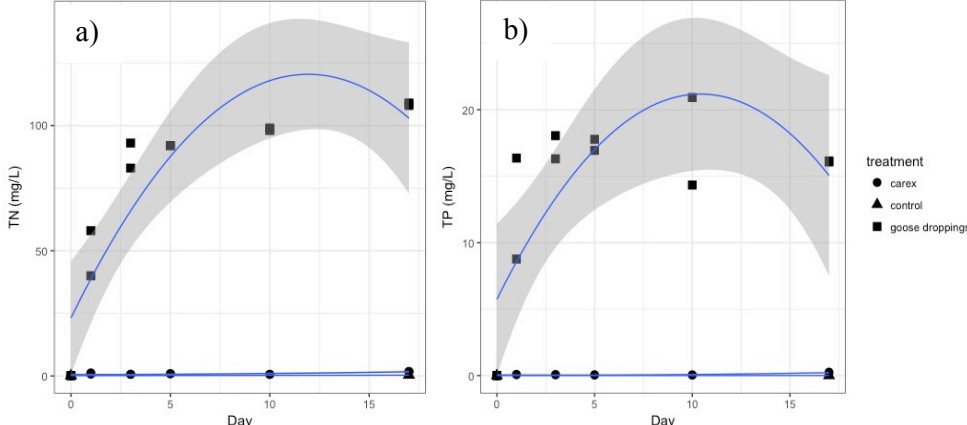

**Figure 3. The cumulative nutrient concentrations released from goose droppings and *Carex* sp. into the water column a) total dissolved nitrogen (TN) and b) total dissolved phosphorus (TP) in mg L$^{-1}$ over the 17-day mesocosm experiment. Dots represent individual samples, on each day duplicate samples were taken. Quadratic function used to fit data with shaded area representing the 95% confidence interval.**





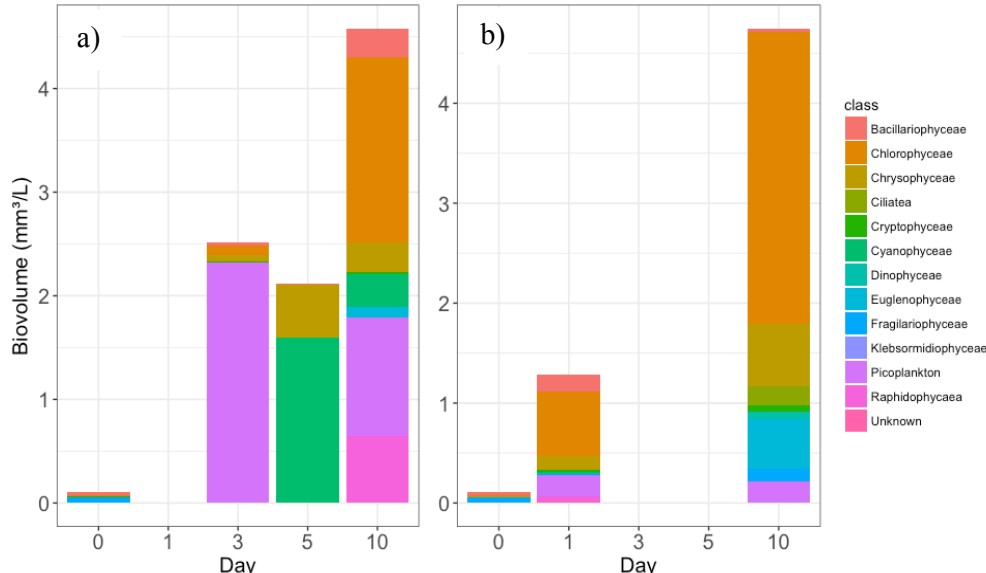

**Figure 4. Change in phytoplankton community composition by biovolume (mm³ L⁻¹) grouped by class, measured over the first 10 days of the mesocosm experiment that used either submerged a) goose droppings or b) *Carex* sp. to stimulate phytoplankton production in lake water. Missing values were due to incomplete preservation of those samples.**

