# Peer review of "Experimental tests of nutrient release and planktonic responses to ornithological eutrophication in Arctic freshwaters"

_Biogeosciences, 2019_

## Referee Comment (RC1) · Anonymous Referee #1 · 26 Jun 2019

This study provides interesting new knowledge on the important role of goose droppings in affecting water quality in Arctic freshwaters. It shows that these droppings have a greater short term effect on water quality than a sedge plant. These results are not unsurprising. The paper is well written and clear. The experimental design is simple and straightforward. The parameters measured are basic water quality parameters, although chlorophyll a concentrations were not measured. The findings from using small containers should not be over-interpreted. These small containers have a high surface area to volume ratio which can become important in terms of biofilm growth on walls. So the effect in the first few days is the most ecological relevant. This limitation should be discussed. Nitrogen versus phosphorus limitation is only relevant

if concentrations are low. Therefore N:P ratios should be with caution. Additionally, N fixing cyanobacteria are only promoted if N concentrations are low, not just because N:P ratios are low. The study did not measure primary production, that is a rate process. So the paper should be explicit that what was measured was accumulation of biovolume.

---

## Referee Comment (RC2) · Anonymous Referee #2 · 10 Jul 2019

In this manuscript Mariash and co-workers investigate the impact of goose faeces on water chemistry and phytoplankton communities in arctic freshwaters. The authors present results from a mesocosm experiment in which goose faeces and Carex leaves were added in different treatments. In the experiment the development of water chemistry (C, P and N) and phytoplankton community were followed. Estimated goose faeces loading rates, water chemistry and chlorophyll data (unpublished data) from a previous study by the authors in the Southhampton area were compared to with results from other studies in USA and Canada.

Given the increasing goose populations in the arctic and the impact they have on the

arctic environment the subject of the manuscript is highly relevant. This is especially true considering that the main focus within this subject so far has been on goose impacts in the terrestrial environment while little attention has been devoted to impacts on the freshwater environment. The manuscript is well written and the language fluent. However, I do have some major comments that detracts from the overall impression of the manuscript.

A major comment concerns the focus of the manuscript as related to the results presented. The title gives the impression that this is a story about goose/bird mediated impact on phytoplankton communities in arctic freshwaters. Furthermore, a large part of the text is devoted to phytoplankton. However, in my mind the strong part of the data presented are the results on changes in water chemistry in response to addition of goose faeces. In fact, it is quite interesting to see how rapid the nutrients from the faeces are released into the water. Due to the limitations outlined below, I do not think that the phytoplankton data presented are substantial enough to back up a main focus on effects on the phytoplankton community.

3 out of 10 (or 8?) phytoplankton samples from the experiment could not be counted due to insufficient fixation. This is a significant part of the samples, especially since the missing samples were from day 1 in the goose faeces treatment and from day 3 and 5 in the Carex treatment, respectively. This made it impossible to compare the phytoplankton response between treatments on day 1, 3 and 5, which is the most important period of the experiment in my mind.

The experimental design is clear. However, a consequence of the phytoplankton sampling, where three replicates (per sampling day/treatment) were pooled, there is just one observation/replicate on phytoplankton biovolume/composition per sampling day/treatment. Considering the focus of the manuscript on phytoplankton responses, it would have been good with more observations/replicates to get an idea about the variation in the phytoplankton response.

It is stated that one of the aims was to measure changes in phytoplankton productivity. In the methods it is explained that phytoplankton production rates were estimated from changes in biovolumes between different sampling days. Thus, it is not the phytoplankton productivity that is measured but phytoplankton biovolumes. In the results section no production estimates are presented. Here, only the changes in the biovolumes are mentioned. I guess, this is due to the lacking phytoplankton samples mentioned above making comparison between treatments impossible throughout most of the period that phytoplankton was sampled in the experiments.

Phytoplankton taxa were identified to genera. Species within the same genera may show different responses to a given environmental change, e.g. eutrophication. Hence, a better taxonomical resolution in the identification would have made it possible also to get an idea about the species specific responses to the experimental treatments.

For reasons outlined above, I do not think the data presented on phytoplankton justifies the focus in the manuscript on phytoplankton response to goose impacts in arctic freshwaters. However, the data presented on the changes in water chemistry in the goose treatment are very interesting. Especially the rapid release of nutrients from the goose faeces brings new insights. I suggest the authors rework the manuscript to "Experimental tests of water chemistry to goose mediated (or ornithological?) eutrophication in Arctic freshwaters", leaving out the phytoplankton data from the manuscript.

Minor comments: In the introduction there is a focus on the nutrient enrichment as the mechanistic explanation for the goose impact on arctic freshwaters. I agree that this is likely an (the most) important mechanism in explaining goose impacts. Still, dispersal effects in the experiments cannot be completely excluded due to the way goose faeces additions were done (no treatment of the faeces was done, e.g. heating, before addition to the experimental beakers). Hence, differences in phytoplankton composition between treatments could also, at least partly, be caused by input of phytoplankton spores/cysts with the goose faeces. This is not a criticism of the experimental setup, but I think this issue should be mentioned in the manuscript (introduction and discussion) as long as the phytoplankton aspect is included in the manuscript. P. 5 l. 15-16: "(for sampling details see Mariash et. al. 2018, Table 3)" instead of "(for sampling details see Mariash et. al. 2018)(Table 3)". P. 5 l. 27-28: Production rates are not included in the results section. P. 5 l. 28: include identification literature. P. 6 l. 4-5: "To model the effect of our treatments on primary productivity, we used a linear mixed effects model implemented in lme4 (Bates et al., 2014)." As far as I can see these results are not included in the results section, likely due to the lacking phytoplankton data. P. 7 l. 8-16: This text passage seems a bit strange due to the lacking phytoplankton data. This also applies to l. 24-26 on p. 7. P. 7 l. 23: delete "(". P. 9. L. 19: "We had expected phytoplankton production to be higher in the goose dropping treatment vs. the Carex treatment". As noted above phytoplankton itself is not measured. Thus, it seems safer to use "phytoplankton biovolume". P. 9 l. 28-32: Also, this text passage seems a bit strange due to the lacking phytoplankton data. Figure 4: it is difficult the distinguish the colors used for some of the phytoplankton classes, e.g. dinophyceae, euglenophyceae and fragilariophyceae. Maybe use different patterns in grey tones instead.

---

## Author Comment (AC1) · 2 Aug 2019

1. This study provides interesting new knowledge on the important role of goose droppings in affecting water quality in Arctic freshwaters. It shows that these droppings have a greater short-term effect on water quality than a sedge plant. These results are not unsurprising. The paper is well written and clear. The experimental design is simple and straightforward. The parameters measured are basic water quality parameters, although chlorophyll-a concentrations were not measured.

Thank you for the positive feedback on several aspects of the manuscript. We underestimated the amount of organic matter in the goose treatments when designing the
experiment, as such several aspects such as chlorophyll-a and primary productivity failed to be realized.

2. The findings from using small containers should not be over-interpreted. These small containers have a high surface area to volume ratio which can become important in terms of biofilm growth on walls. So the effect in the first few days is the most ecological relevant. This limitation should be discussed. We have included an additional sentences to P.9 to acknowledge the potential container effects: "Our experimental design was not without issues: more replicates would have made the results clearer, and the use of small containers has the potential to contribute technique-related artefacts (e.g., biofilm growth, altered physiochemical conditions, and species interactions due to container area-to-volume relationship; Liber et al., 2007), effects that we attempted to mitigate through the use of s short experimental duration. Despite these caveats, the responses in the phytoplankton communities were pronounced."

3. Nitrogen versus phosphorus limitation is only relevant if concentrations are low. Therefore N:P ratios should be with caution. Additionally, N fixing cyanobacteria are only promoted if N concentrations are low, not just because N:P ratios are low.

On P. 10 of the Discussion we have a sentence that highlights when low nitrogen concentrations promote cyanobacteria, not just low N:P ratios. "This is an environmental concern from a water quality perspective, because when N is limiting, N2-fixing cyanobacteria are competitively favoured (Guildford and Hecky, 2000; Schindler et al., 2008)" To decrease the emphasizes on the N:P ratios in this same paragraph, we have removed this sentence ("In the wintering grounds with highest goose densities, TN:TP ratios of the waterbodies had a mean of 15 (Kitchell et al., 1999), indicating N- limitation. While the" Lastly we adjusted the last sentence to include that we have both low N and decreasing N:P ratios. "The wetlands across Southampton Island have relatively low nitrogen concentration and TN:TP ratios of approximately 30 (Mariash et al. 2018), on par with other shallow Arctic freshwaters (Rautio et al., 2011), there is an indication that these wetlands are becoming more N-limited with decreasing TN:TP

ratios (Mariash et al., 2018)."

4. The study did not measure primary production, that is a rate process. So the paper should be explicit that what was measured was accumulation of biovolume. We are now more explicit throughout the manuscript using biovolume and not the rate of primary productivity. Also Reviewer 2 made several suggestions in this regard, please refer to those comments for specific amendments.
* * *

---

## Author Comment (AC2) · 2 Aug 2019

1. In this manuscript Mariash and co-workers investigate the impact of goose faeces on water chemistry and phytoplankton communities in arctic freshwaters. The authors present results from a mesocosm experiment in which goose faeces and Carex leaves were added in different treatments. In the experiment the development of water chemistry (C, P and N) and phytoplankton community were followed. Estimated goose faeces loading rates, water chemistry and chlorophyll data (unpublished data) from a previous study by the authors in the Southhampton area were compared to with results from other studies in USA and Canada. Given the increasing goose populations in the

arctic and the impact they have on the arctic environment the subject of the manuscript is highly relevant. This is especially true considering that the main focus within this subject so far has been on goose impacts in the terrestrial environment while little attention has been devoted to impacts on the freshwater environment. The manuscript is well written and the language fluent.

We appreciate the reviewer's recognition of the value and relevance of the data that we report.

2. A major comment concerns the focus of the manuscript as related to the results presented. The title gives the impression that this is a story about goose/bird mediated impact on phytoplankton communities in arctic freshwaters. Furthermore, a large part of the text is devoted to phytoplankton. However, in my mind the strong part of the data presented are the results on changes in water chemistry in response to addition of goose faeces. In fact, it is quite interesting to see how rapid the nutrients from the faeces are released into the water. Due to the limitations outlined below, I do not think that the phytoplankton data presented are substantial enough to back up a main focus on effects on the phytoplankton community.

We will modify the manuscript using the reviewer's suggested improvements to shift the focus more to the water chemistry and less emphasis on the phytoplankton results.

3. 3 out of 10 (or 8?) phytoplankton samples from the experiment could not be counted due to insufficient fixation. This is a significant part of the samples, especially since the missing samples were from day 1 in the goose faeces treatment and from day 3 and 5 in the Carex treatment, respectively. This made it impossible to compare the phytoplankton response between treatments on day 1, 3 and 5, which is the most important period of the experiment in my mind.

We agree with the reviewer that the loss of three critical samples does impact the extent to which we can draw conclusions regarding changes to the phytoplankton community. However, the relative comparison between the treatments is still valuable as they indicate the magnitude and compositional difference between goose droppings and carex treatments.

4. The experimental design is clear. However, a consequence of the phytoplankton sampling, where three replicates (per sampling day/treatment) were pooled, there is just one observation/replicate on phytoplankton biovolume/composition per sampling day/treatment. Considering the focus of the manuscript on phytoplankton responses, it would have been good with more observations/replicates to get an idea about the variation in the phytoplankton response.

We agree with the reviewer and appreciate the recommendation. Including more replicates would have allowed for more variability to be seen within the phytoplankton response between and within treatments. In the future, our experimental set up will include higher sample volumes to accommodate for replication and higher concentrations of fixative to compensate for the organic material in the samples.

5. It is stated that one of the aims was to measure changes in phytoplankton productivity. In the methods it is explained that phytoplankton production rates were estimated from changes in biovolumes between different sampling days. Thus, it is not the phytoplankton productivity that is measured but phytoplankton biovolumes. In the results section no production estimates are presented. Here, only the changes in the biovolumes are mentioned. I guess, this is due to the lacking phytoplankton samples mentioned above making comparison between treatments impossible throughout most of the period that phytoplankton was sampled in the experiments.

Yes, as mentioned above, the loss of samples made direct comparisons between treatments challenging. We will edit the manuscript to discuss "phytoplankton response" rather than "phytoplankton productivity" as we did not present true primary productivity results. We have changed the aim from "phytoplankton production" to changes in "phytoplankton biovolume".

6. Phytoplankton taxa were identified to genera. Species within the same genera may

show different responses to a given environmental change, e.g. eutrophication. Hence, a better taxonomical resolution in the identification would have made it possible also to get an idea about the species-specific responses to the experimental treatments. For reasons outlined above, I do not think the data presented on phytoplankton justifies the focus in the manuscript on phytoplankton response to goose impacts in arctic freshwaters.

Although species-specific response would have been best, identification to genera still provides a good comparison between treatments. We have taken the reviewer's suggestions on how to better present the phytoplankton results throughout the manuscript.

7. However, the data presented on the changes in water chemistry in the goose treatment are very interesting. Especially the rapid release of nutrients from the goose faeces brings new insights. I suggest the authors rework the manuscript to "Experimental tests of water chemistry to goose mediated (or ornithological?) eutrophication in Arctic freshwaters", leaving out the phytoplankton data from the manuscript.

We appreciate the reviewer's suggestion and will change the title to include "water chemistry and planktonic response" to better reflect the water chemistry focus of the manuscript. We will rework the manuscript to focus on the water chemistry results more; however, the phytoplankton results are a valuable part of the data demonstrating how the biotic component reacts to the changes in chemistry. We will keep the phytoplankton data in the manuscript using amendments suggested from both reviewers to shift the focus towards the water chemistry results.

8. Minor comments: In the introduction there is a focus on the nutrient enrichment as the mechanistic explanation for the goose impact on arctic freshwaters. I agree that this is likely an (the most) important mechanism in explaining goose impacts. Still, dispersal effects in the experiments cannot be completely excluded due to the way goose faeces additions were done (no treatment of the faeces was done, e.g. heating, before addition to the experimental beakers). Hence, differences in phytoplankton composition between treatments could also, at least partly, be caused by input of phytoplankton spores/cysts with the goose faeces. This is not a criticism of the experimental setup, but I think this issue should be mentioned in the manuscript (introduction and discussion) as long as the phytoplankton aspect is included in the manuscript.

We agree that the main mechanism of goose impacts is through nutrient inputs, but species dispersal is also possible. We have added a couple of sentences to introduction and discussion to include this aspect.

In Introduction P. 2. Inserted "Birds can also act as vectors for the dispersal of plants, phytoplankton, and zooplankton, when propagules are spread through their faeces eggs (Figuerola and Green, 2002; Hessen et al., 2019).

In discussion, P. 10. Amended the 3rd paragraph on P10, now includes: "Geese are therefore acting as biovectors on the landscape, consuming large amounts of terrestrial nutrients bound in vegetation and excreting these nutrients in form that is bioavailable for freshwater ecosystems. Goose faeces could also contribute to the dispersal of aquatic species, altering aquatic communities in this direct manner (Figuerola and Green, 2002). Tested phytoplankton species were not viable under cultured conditions once passed through waterbirds (Atkinson 1980), however tests of this mechanism have not yet been carried out for geese in the Arctic."

9. P. 5 l. 15-16: "(for sampling details see Mariash et. al. 2018, Table 3)" instead of "(for sampling details see Mariash et. al. 2018)(Table 3)". Added a semi colon instead of the double bracket. 10. P. 5 l. 27-28: Production rates are not included in the results section. Correct, this sentence has now been removed. 11. P. 5 l. 28: include identification literature. Now in the methods: "Phytoplankton taxonomy relyed on the following literature Cox, 1996; Millebrand et al, 1999; Whitton and Brook, 2002; Komárek and Anagnostidis, 2000; Guiry and Guiry, 2017; Taylor and Archibald, 2007; and Wehr and Kociolek, 2015."

12. P. 6 l. 4-5: "To model the effect of our treatments on primary productivity, we used

a linear mixed effects model implemented in lme4 (Bates et al., 2014)." As far as I can see these results are not included in the results section, likely due to the lacking phytoplankton data. True, these sentences correspond to data that we no longer present in this paper, and thus these sentences should be deleted.

13. P.7 l. 8-16: This text passage seems a bit strange due to the lacking phytoplankton data. This also applies to l. 24-26 on p. 7. We have amended to read: "While the biovolumes were similar, the phytoplankton communities were different between treatments." 14. P. 7 l. 23: delete "(". P. 9. L. 19: "We had expected phytoplankton production to be higher in the goose dropping treatment vs. the Carex treatment". As noted above phytoplankton itself is not measured. Thus, it seems safer to use "phytoplankton biovolume". We have changes "phytoplankton production" to phytoplankton biovolumes" 15. P. 9 l. 28-32: Also, this text passage seems a bit strange due to the lacking phytoplankton data. We have amended the sentence to be more cautious and include suggestions from both reviewer's. The paragraph now reads: "Community composition of the phytoplankton responded to increased nutrient availability in both treatments. Our experimental design was not without issues: more replicates would have made the results clearer, and the use of small containers has the potential to contribute technique-related artefacts (e.g., biofilm growth, altered physiochemical conditions, and species interactions due to container area-to-volume relationship; Liber et al., 2007), effects that we attempted to mitigate through the use of s short experimental duration. Despite these caveats, the responses in the phytoplankton communities were pronounced."

16. Figure 4: it is difficult the distinguish the colors used for some of the phytoplankton classes, e.g. dinophyceae, euglenophyceae and fragilariophyceae. Maybe use different patterns in grey tones instead. With 14 classes it is difficult to get a lot of distinction when the layers are so small. Since it is for online publication, we prefer to use color rather than grey-scale. We will custom build the color pallet to improve the distinction between colors.

---

## Author Response (AR2)

**Associate Editor Decision: Reconsider after major revisions** (30 Sep 2019) by Perran Cook

Comments to the Author:
Dear Dr. Mariash
Thank you for your response to the reviewers comments. I note reviewer 2 still has concerns about the presentation of the phytoplankton data, but is supportive of the publication of the nutrient data. The main issue is the lack of replication which means that it is impossible to say whether there was indeed a statistically significant effect on the phytoplankton community. I am, however of the view, that some information (especially hard to collect data) is better than no information, particularly when presented alongside other data and that these results can remain in the manuscript provided that the caveats are clearly outlined and that your statements and title reflect this. My key suggestion for changes include:

Thank you to both the reviewer for another round of comments and to the editor for seeing the importance of keeping the phytoplankton data in the manuscript. We agree that it is important to link the chemical changes with the potential biological changes.

1. Title – remove all reference to phytoplankton response

Removed all reference to phytoplankton in title. New title "Experimental tests of water chemistry response to ornithological eutrophication: biological implications in Arctic freshwaters"

2. Discussion – You need to state clearly upfront that no statistically significant statements can be made about the differences between the treatments (for example you cannot say there was a pronounced response). I think it is ok to then discuss in a short paragraph how the generic differences relate to previous work on eutrophication (ideally in the similar cold waters) and the nutrient concentrations. I note your observations for picoplankton seem to be inconsistent with https://pdfs.semanticscholar.org/ba64/353992834361aa34fc9e8badd9b00bc29ff2.pdf

P.10Line 5. We have removed "pronounced response" and instead added a sentence that clearly states the phytoplankton response is an indication not a certainty of phytoplankton response to treatments.
"Due to these limiting factors, the experiment can best be used to show the biological implications of the released nutrients and potential phytoplankton response."
In a few other places, ex. P.10.Line 30, we added the word "likely".

Figure 4 – a and b are swapped? At the moment the figure shows the opposite of what you state in the text

Also, please refer to 4a and b in alphabetical order in the results.

Fixed Fig. 4 letter and made sure the letter is in alphabetical order in text. Carex is Fig. 4a and goose droppings is Fig. 4b.

**Reviewer**

**Suggestions for revision or reasons for rejection (will be published if the paper is accepted for final publication)**

In the revised ms the authors have incorporated the minor comments I had. My major concern was the focus on phytoplankton when considering the limitations of the phytoplankton data presented. I recommended to change the title and rework the ms, leaving out the phytoplankton data. The authors did change the title, but they kept the phytoplankton data. In my mind there is still a major focus on the effects on the phytoplankton community in the revised ms. I agree with the authors that the phytoplankton response is interesting. However, the limitations of the phytoplankton data listed in my initial comments detracts from the overall quality of the ms. Still, I think the development of the water chemistry is very interesting and could make a nice story. Hence, I maintain that the ms should be revised focusing on this part and leaving out the phytoplankton data.

As per suggestion of reviewer and editor we have changed the title removing the word "phytoplankton".

New title is: "Experimental tests of water chemistry response to ornithological eutrophication: biological implications in Arctic freshwaters"

Based on the editor's suggestion we have kept the phytoplankton in the manuscript but made the discuss paragraph clearer with the limitations of the experiment and conclusions that can be drawn.

Minor comments

P. 6, l.3: ")" missing after mL.- added ")" after mL

p. 6. L. 8: ")" missing after clippings- added ) after clippings

Figure 2: "a)", "b)", "c)" and "d)" mentioned in the figure text is missing in the figure. Also the upper right panel is shifted towards right.- fully edited fig 2 to high resolution including alignment, font, size and colour.

Figure 4: The lettering in the figure has been switched.- edited Fig. 4 to high resolution, larger font size in legend, and corrected the letters of on panels.

Edited fig 3 as well to high resolution and adjusted panel size and legend font size.

[revised manuscript text omitted]